



# 1750 years of hydrological change in southern Australia: a bivalve oxygen isotope record from the Coorong Lagoon

Briony K. Chamberlayne[1], Jonathan J. Tyler[1], Deborah Haynes[1,2], Yuexiao Shao[1], John Tibby[2], Bronwyn M. Gillanders[3]

[1] Department of Earth Sciences and Sprigg Geobiology Centre, The University of Adelaide, Adelaide, 5005, Australia
[2] Department of Geography, Environment and Population, The University of Adelaide, Adelaide, 5005, Australia
[3] Southern Seas Ecology Laboratories and the Environment Institute, School of Biological Sciences, The University of Adelaide, Adelaide, 5005, Australia

*Correspondence to*: Jonathan J. Tyler (Jonathan.tyler@adelaide.edu.au)

**Abstract.** Multi-centennial records of past hydroclimate change are essential to understanding the resilience of aquatic ecosystems to climatic events, in addition to guiding conservation and restoration efforts. Such data are also crucial for examining the long-term controls over regional hydroclimate, and the inherent variability in extreme droughts and floods. Here, we present a 1750-year record of hydroclimate variability in The Coorong South Lagoon, South Australia, part of an internationally significant wetland system at the mouth of Australia's largest river, the Murray River. Oxygen isotope ratios were measured in *Arthritica helmsi* bivalve shells preserved in sediments. The oxygen isotope record shows periods of persistent low and high moisture balance from ~500–1050 years and ~1300–1800, respectively, which is consistent with other hydroclimate reconstructions from the region. The range of oxygen isotope values in the sedimentary shells do not differ significantly from the estimated range of modern specimens from the present day lagoon. These data suggest that the restricted and highly evaporated modern day conditions are not markedly different to the pre-impacted state over the last 1750 years, although the absence of *A. helmsi* in the contemporary lagoon is likely a response to increased salinity, nutrient loading, and anoxia during the last century. These insights are potentially useful both in guiding management efforts in the currently degraded South Lagoon, as well as for understanding long term water resource availability in the region.

## 1 Introduction

Multi-decadal to centennial records of past hydroclimate variability are crucial for understanding long term climate drivers, for calibrating and validating climate models, for assessing hydroclimate sensitivity to external drivers and for estimating the probability of multi-decadal climate extremes (Ault et al., 2014; Barr et al., 2014, 2019; Cook et al., 2016; Ho et al., 2015; Kiem et al., 2016; Verdon and Franks, 2007). Such data are particularly important for the economic and environmental sustainability of the arid and semi-arid regions on the continents of the Southern Hemisphere. In addition, the inherent links between Southern Hemisphere hydroclimate and variability in the Southern Westerly Winds (SWW) affect oceanographic circulation and carbon cycling between the ocean and atmosphere. Recently, the vegetation and soils in semi-arid landscapes





of Australia and Africa have been shown to be major carbon sinks (Haverd et al., 2013; Ma et al., 2016; Poulter et al., 2014), placing further demand upon long term hydroclimate data in these regions.

To date, decadal scale records of hydroclimate variability in southern Australia covering the last two millennia are relatively rare, mostly due to a scarcity of sufficiently long tree ring chronologies and the geographically narrow distribution of lake and speleothem archives (Dixon et al., 2017; Jansen et al., 2007; Neukom and Gergis, 2011). A recent review of hydroclimate records for the past 2000 years in Australasia identified only nine 'high-quality' records from Australia, i.e. meeting the criteria for the PAGES2k network (Dixon et al., 2017). Alternative reconstructions of Australian hydroclimate variability have utilised
distal archives, such as Antarctic ice cores, New Zealand tree rings and Chinese speleothems (McGowan et al., 2009; Palmer et al., 2015; Tozer et al., 2016; Vance et al., 2013) however major uncertainties exist concerning the stationarity of climate teleconnections when applied beyond the calibration period and geographic range (Dätwyler et al., 2018; Gallant et al., 2013). The relatively low number of high-quality records in Australia limits our understanding of the long-term influence of remote drivers of rainfall, namely the ocean-atmosphere interactions El Niño Southern Oscillation (ENSO), Interdecadal Pacific
Oscillation (IPO), Indian Ocean Dipole (IOD) and Southern Annular Mode (SAM) indices (Gallant et al., 2012; Risbey et al., 2009). A better understanding of natural, low frequency climate variability associated with these climate drivers, along with the reconstruction of decadal to multi-decadal climate extremes such as drought or flood, could assist in placing recent events into context and aid in assessing the risks associated with future climate events (e.g. O'Donnell et al., 2021). This is especially relevant for economically important estuaries with populous catchments that are highly sensitive to climatic extremes (e.g.
Scanes et al., 2020).

The Coorong Lagoon in South Australia is one such estuarine ecosystem for which there is limited knowledge of past hydroclimate variability (Kingsford et al., 2011; Reeves et al., 2015). The Coorong, along with Lakes Alexandrina and Albert (the Lower Lakes) form a large wetland at the mouth of the Murray River, which along with the Darling River drains the >1,000,000 km2 Murray-Darling Basin (MDB). The MDB is Australia's most important agricultural region, providing around
40% of Australia's produce, while the region generally is of great economic, cultural and environmental importance to Australia (MDBA, 2021). The Coorong, along with 15 other wetlands in the MDB, was recognised as a wetland of global significance by inclusion in the Ramsar Convention for Globally Significant Wetlands (Department of Agriculture, Water and the Environment, 2021) primarily due to its biodiversity, especially for waterbird species (Paton et al., 2009). Furthermore, The Coorong and Lower Lakes are a culturally important site for the Ngarrindjeri people (Ngarrindjeri Nation & Hemming,
2019). In recent years, the condition of The Coorong Lagoon has deteriorated, with rising salinity, eutrophication and declining biodiversity all linked to declining river discharge due to agricultural and consumptive water extraction (Dittmann et al., 2019; Kingsford et al., 2011; Mosley et al., 2020). Current management efforts are focussed on increasing system flushing and reversing eutrophication (Mosley et al., 2020) including consideration of proposals to open the South Lagoon of The Coorong to the Indian Ocean. However, as an inverse estuary, the South Lagoon of The Coorong is a unique environmental system and

despite its proximity to the ocean, its natural (pre-impacted) hydrological state is not well understood (Reeves et al., 2015). In this respect, palaeo-environmental studies can provide context regarding the long-term condition of The Coorong, including the range of and resilience to natural hydroclimate variability (Saunders and Taffs, 2009).

Several studies have been undertaken to understand the palaeo-environment of The Coorong Lagoon. Many of these investigated fossil invertebrate and algal assemblages (Dick et al., 2011; Fluin et al., 2007; Krull et al., 2009; Lower et al.,
2013; Reeves et al., 2015) and have concluded that the current state of the wetland is unusual in its history. Moreover, many of the dominant species found in the assemblages have broad environmental tolerances and hence gave rise to ambiguity in the interpretations. Other studies have included investigations of the carbon, nitrogen and hydrogen isotope composition of organic matter as well as lipid biomarkers as proxies for variations in the source of organic matter and salinity (McKirdy et al., 2010; Tulipani et al., 2014). These methods have found that the North and South Lagoons have been separate chemical
systems for millennia (McKirdy et al., 2010) although both lagoons have experienced increased salinity levels more recently (McKirdy et al., 2010; Tulipani et al., 2014). Modern sediments have been well dated using Lead-210 (210Pb), fallout product Caesium-137 (137Cs), Plutonium (238/239/240Pu) and the first arrival of exotic Pinus radiata pollen (Fluin et al. 2007; Krull et al., 2009). However, several previous studies have encountered challenges related to radiocarbon dating uncertainty, largely due to a relatively small number of dates, which make it difficult to place long term changes into a regional context (e.g
McKirdy et al., 2010).

Here, we use the geochemistry of the bivalve *Arthritica helmsi*, which are abundantly preserved in Coorong South Lagoon sediments, to add to the growing body of research investigating the palaeoenvironmental history of this significant wetland. A modern study of oxygen isotope fractionation in *A. helmsi* demonstrated the sensitivity of bivalve geochemistry from this species to environmental change (Chamberlayne et al., 2021). In this study we analyse oxygen isotope ratios in shells preserved
in a sediment core from the South Lagoon to (i) assess the drivers of hydrological change in the context of regional climate variability; and (ii) understand hydrological change in the South Lagoon.

## 2 Methods

### 2.1 Site description

The Coorong Lagoon and the Goolwa Channel comprise the River Murray estuary (Figure 1). The Coorong is separated from
the Indian Ocean by Holocene dunes (Murray-Wallace, 2018) through which the Murray Mouth provides the drainage for the Murray- Darling River system. Formally speaking, The Coorong Lagoon is a single entity; however, the constriction at Parnka Point effectively divides The Coorong into two separate basins. While the basins do not have specific names, from a hydrological, geochemical and ecological perspective they are quite distinct from each other, and recent scientific and management efforts have had a specific focus on the southern basin. As a consequence, as is the case with several other studies



on The Coorong, this study adopts the informal nomenclature of the North Lagoon and South Lagoons (Figure 1). The Coorong is characterised by a north-south salinity gradient of fresh to hypersaline waters (Gillanders and Munro, 2012; Shao et al., 2021). A series of barrages constructed in the 1930s to restrict marine incursions and retain freshwater in Lake Alexandrina now regulate freshwater discharges from Lake Alexandrina into The Coorong and Murray Mouth (Maheshwari et al., 1995). Currently, water in the North Lagoon is recharged by outflow from Lake Alexandrina and inflows from the Indian Ocean, while the South Lagoon is primarily recharged by direct rainfall, groundwater and continental run-off from wetlands in the southeast of the state (Chamberlayne et al., 2021; Shao et al., 2018)

## 2.2 Sample collection and preparation

Core C18 was extracted from the South Lagoon (Figure 1) in 2005 using a percussion corer with 50 mm internal diameter PVC pipe (Chambers and Cameron, 2001) and stored at 4°C in refrigerated facilities at The University of Adelaide. The 130 cm core was sliced longitudinally and one half was sampled for diatom and preliminary radiocarbon analyses (Krull et al., 2009). The remaining half of the core was then cut into 1 cm intervals and freeze dried. *Arthritica helmsi* shells were picked for analysis from the dried sediment and cleaned using a brush and distilled water before being oven dried at 40 °C for 12 hours.

## 2.3 Chronology

The chronology for the C18 record was established via a combination of accelerator mass spectrometry (AMS [14]C) radiocarbon dating and the first appearance of *Pinus radiata* pollen. Seventeen [14]C dates were obtained from bivalve shells sampled throughout the core (Table S1). The exotic *Pinus radiata* pollen was first introduced to Australia by Europeans and its detection in sediments can be used as a chronostratigraphic marker for post-colonial expansion of European settlement and agriculture in the region (Tibby et al., 2003). [210]Pb and [137]Cs dating of adjacent cores in The Coorong Lagoon indicates that the first detection of *Pinus* is representative of ~1955 CE (Krull et al., 2009). In core C18, *Pinus* first appears at 9 cm depth (Krull et al., 2009).

To assess the presence of 'old carbon' from groundwater which has affected radiocarbon dating in other southeast Australian lakes (Barr et al., 2014; Gouramanis et al., 2010; Wilkins et al., 2012), the [14]C age for the depth corresponding to *Pinus* arrival was compared to a [210]Pb inferred [14]C age ([14]C$_{210Pb}$), with uncertainty, following Barr et al. (2014). This was achieved by projecting a probability density curve based on the mean [210]Pb age and standard deviations derived from Krull et al. (2009) against the Southern Hemisphere [14]C calibration curve (Hogg et al., 2020). The mean [14]C age within the error bracket was used to estimate [14]C$_{210Pb}$, weighted by its probability. The reservoir age was then estimated by subtracting [14]C$_{210Pb}$ from measured [14]C, and the resulting uncertainty was calculated using following equation:

$$\sigma = \sqrt{\sigma_{res}^2 + \sigma_{atm}^2} \tag{1}$$



where $\sigma_{res}$ and $\sigma_{atm}$ are the errors associated with the [14]C age of the reservoir-derived and atmosphere-derived objects respectively (Soulet, 2015). This old carbon effect was then subtracted from all [14]C measurements from core C18 prior to calibration with the Southern Hemisphere calibration curve (Hogg et al., 2020) using the program Bacon (Blaauw and Christen, 2011) in RStudio (R Core Team, 2021).

**2.4 Stable isotope analysis**

In order to obtain a representative sample of the past environment, multiple individuals were combined for each analysis. Five pre-cleaned shells, collected in the 1 cm sample intervals throughout core C18 were crushed to a fine powder using an agate mortar and pestle. A sub-sample of approximately 100 μg from this carbonate powder was used for stable isotope analysis. In addition, to assess the degree of intra-sample variability, multiple individual shells of *A. helmsi* were also analysed from samples taken at five depths, namely 11 cm (n=30 shells), 26 cm (n=29), 57 cm (n=27), 92 cm (n=30) and 119 cm (n=27).

Data from these intervals were integrated into final record by calculating the weighted mean of the bulk sample and replicate samples from particular depths. Samples were flushed with helium and then acidified with 105% phosphoric acid at 70°C using a Nu Instruments GasPrep device, before isotope analysis of the resulting $CO_2$ with a Nu Instruments Horizon isotope ratio mass spectrometer in continuous flow mode. Laboratory standards ANU-P3 ($\delta^{18}O = -0.3$‰) and UAC-1 ($\delta^{18}O = -18.4$‰) accounted for 27 of each 100 analyses, the precision was better than 0.12 ‰ for $\delta^{18}O$ based on replicate analyses of these

standards. Oxygen isotope data are reported as $\delta^{18}O_s$ relative to Vienna Pee Dee Belemnite (VPDB) standard using the standard delta (δ) notation in parts per thousand (‰):

$$\delta = \left(\frac{R_{sample} - R_{standard}}{R_{standard}}\right) \times 1000 \tag{2}$$

where R is the isotope ratio ($^{18}O/^{16}O$).

**2.5 Comparison of oxygen isotope ratios in core C18 and modern *Arthritica helmsi***

The oxygen isotope ratios measured on *A. helmsi* shells in core C18 were compared to both measured and predicted oxygen isotope ratios of shell carbonate in the modern North and South Lagoons. Oxygen isotope ratios for live collected individuals of *A. helmsi* from the North Lagoon were sourced from Chamberlayne et al. (2021) who measured $\delta^{18}O$ on shells from five locations throughout the North Lagoon from 2016–2018. As there are currently no living populations of *A. helmsi* in the South Lagoon (Dittmann et al., 2019), the oxygen isotope ratio of shell carbonate was predicted using measured temperature and

oxygen isotope ratios in water reported by Chamberlayne et al. (2021). Using this approach, $\delta^{18}O_s$ predictions were made for hypothetical shell populations at Parnka Point and Jack Point using the temperature-dependent oxygen isotope fractionation equation developed for *A. helmsi* by Chamberlayne et al. (2021):

$$T \,(°C) = (21.39 \pm 0.45) - (4.43 \pm 0.38) * (\delta^{18}O_{shell} - \delta^{18}O_{water}) \tag{3}$$



To determine whether the oxygen isotope compositions varied between sites, we first tested if the data sets were normally
distributed using a Shapiro-Wilk test. As the data from core C18 were not normally distributed (w = 0.977, p = 0.037),
Wilcoxon μ-tests were used to compare groups.

## 3 Results

### 3.1 Chronology

The age offset due to 'old carbon' in the South Lagoon was calculated by comparing the $^{14}$C age of bivalves from the depth
corresponding with the first occurrence of *Pinus* pollen in core C18. In other South Lagoon cores, alpha and gamma
spectrometry-based $^{210}$Pb analyses indicate that the first occurrence of *Pinus* in the sediments dates to 1955 ± 5 CE (-5 ± 5
years BP; Fluin et al., 2007; Krull et al., 2009). This $^{210}$Pb derived age was estimated to equate to 169 ± 10 $^{14}$C years BP which
was subtracted from the equivalent measured radiocarbon age (985 ± 30 $^{14}$C years BP) to produce an age offset of 816 ± 32
$^{14}$C years BP. The results of AMS radiocarbon dating are summarised in Table S1. The age-depth model produced following
the application of this offset to measured radiocarbon ages is shown in Figure 2. Three samples were found to be more than
500 years older than the above or below samples and were therefore deemed to be outliers and excluded from the age-depth
model (Table S1). The weighted mean basal age for core C18 is 1755 cal yr BP (195 CE).

### 3.2 Oxygen isotope ratios

The $\delta^{18}O_s$ values measured from each centimetre of core C18 (n= 123) ranged from 2.86 ‰ at 104 cm to 5.20 ‰ at 61 cm with
a mean of 3.95 ‰ (Figure 3a). $\delta^{18}O_s$ values were above average from ~269–439, 529–1065, 1218–1321 and 1789–2005 CE
and below average for the time periods ~195–275, 439–529, 1065–1218 and 1299–1789 CE (Figure 3a). The oxygen isotope
compositions of multiple individual *A. helmsi* shells from five intervals showed high variability (Figure 3b). The median $\delta^{18}O_s$
ranged from 4.61 ± 0.69 ‰ at 119 cm (300 CE) to 3.84 ± 0.64 ‰ at 57 cm (1321 CE; Figure 3b). Excluding outliers, the range
of values measured was highest at 119 cm (2.66 ‰; 300 CE) and lowest at 92 cm (0.99 ‰; 631 CE).
A Wilcoxon test indicated that the distribution of oxygen isotope values measured on aggregate samples of shells of *A. helmsi*
extracted from core C18 were not significantly different from the $\delta^{18}O_s$ values predicted by the modern water $\delta^{18}O$ and
temperature at Jacks Point (p = 0.15) and Parnka Point (p = 0.18) in the South Lagoon (Figure 5a). The range of $\delta^{18}O_s$ values
measured from core C18 was, however, narrower than that predicted for modern carbonates. When the range of measured
$\delta^{18}O_s$ values from the North Lagoon was compared to the values from core C18, the two populations were significantly different
(p < 0.001; Figure 5b). To assess sampling bias, oxygen isotope values for individual *A. helmsi* shells were compared to the
$\delta^{18}O_s$ values predicted for the modern South Lagoon and the range of measured $\delta^{18}O_s$ values from the modern North Lagoon.
In agreement with the aggregate $\delta^{18}O_s$ samples, individual $\delta^{18}O_s$ values are distinctly different from the modern North Lagoon,
yet consistent within the range of estimated $\delta^{18}O_s$ values for the South Lagoon (Figure S1).





## 4 Discussion

Proxy records of hydroclimate are required to understand variability extending past Australia's short instrumental climate record. This study presents variation in the oxygen isotope ratios of shells of *A. helmsi* spanning the last ~1750 years in the South Lagoon of The Coorong. This record is constrained by 17 radiocarbon dates allowing accurate identification of the timing of hydrological change, albeit subject to the need to correct radiocarbon dates for a substantial 'old carbon' effect. Here, we interpret the record in context of previous studies of Coorong palaeohydrology, in addition to regional hydroclimate

reconstructions. Furthermore, the possible drivers of hydrological change in the Coorong's South Lagoon are considered.

### 4.1 Interpreting $\delta^{18}O_s$ in South Lagoon sediments

The processes that control the fractionation of oxygen in the carbonate of *A. helmsi* were examined in a previous study of modern populations in The Coorong Lagoon (Chamberlayne et al. 2021). A temperature dependent fractionation equation was developed in that study which determined that an increase in $\delta^{18}O_s$ by 1 ‰ corresponds to a decrease in temperature by ~4.43

°C (Chamberlayne et al., 2021; Equation 3). Alternatively, a change in water temperature by 1 °C would be expressed as a ~0.23 ‰ change in the oxygen isotope value of *A. helmsi*. As this study sampled multiple shells per sample to reduce seasonal and inter-annual bias in the signal captured by the oxygen isotope composition of shells, we interpret this record to be more heavily influenced by the isotopic composition of water rather than temperature. This interpretation is based, in part, by the relatively low variance in regional temperature reconstructions for the last 2000 years (Kaufman et al., 2020) whilst

acknowledging that mainland Australia remains very poorly represented by global temperature reconstructions for the recent millennia. The oxygen isotopic composition of modern Coorong waters showed a positive linear relationship to salinity, demonstrating an overriding dominance of evaporation over isotopic variability within the lagoon, although the slope and strength of this relationship varied spatially (Chamberlayne et al., 2021). As such, the oxygen isotope record developed in this study is primarily interpreted as reflecting the inflow:evaporation (I:E) water balance of the South Lagoon.

### 4.2 Palaeohydrology of The Coorong Lagoon and south-eastern Australia

The oxygen isotope record from *A. helmsi* implies a relatively stable hydrological state over the last ~1750 years, superimposed by periods of both enhanced and reduced I:E (Figure 3a). In particular, between 500-1050 CE, higher $\delta^{18}O_s$ is interpreted to reflect a period of prolonged, relatively low I:E (Figure 3a). Oxygen isotope values measured from individual shells at A relatively brief period of low $\delta^{18}O_s$ occurred between ~1050–1200 CE, prior to another period of high $\delta^{18}O_s$ between 1200–

1300 CE (Figure 3a). This overall period of relatively dry climate largely coincides with records from western Victoria, approximately 300–500 km south-east of The Coorong (Barr et al., 2014; Dixon et al., 2019; Tyler et al., 2015; Wilkins et al., 2013). Two diatom-inferred conductivity (salinity) records from Lake Surprise and Lake Elingamite indicate both drier and more variable climate during this period (Figure 4; Barr et al. 2014). Diatom-based ecological indicators, such as a greater



proportion of shallow water diatom taxa suggestive of lower lake levels at these and two other lakes (Tower Hill and Lake
Purrumbete) also indicate dryer climates prior to 1300 AD (Barr et al. 2014; Tyler et al. 2015). Warmer temperatures and
lower lake-levels were also inferred from the increased frequency of aragonite laminae in the sediments of Lake Keilambete
and Lake Gnotuk in western Victoria (Figure 4; Wilkins et al., 2013). Off the coast of South Australia, alkenone-derived sea
surface temperatures (SST) in the Great Australian Bight suggest a warm water anomaly from 650 to 950 CE, despite an
overall cooling trend in the late Holocene (Perner et al., 2018). Chronological uncertainties preclude the precise alignment of
these records, however the broad agreement suggests that a period of relatively dry climate occurred between ~ 500 to 1200
CE across the south-eastern Australia (Figure 4).

Excursions in compound specific hydrogen isotope ratios of *n*-alkanes in South Lagoon sediments, alongside diatom-inferred
reductions in salinity, suggested two major freshening events at ~1100 and ~1410 CE (McKirdy et al., 2010). The timing of
these events, which were attributed to increased surface runoff and groundwater inflow (McKirdy et al., 2010) are broadly
coincident with negative excursions in $\delta^{18}O_s$ derived in this study, although additional freshening events are inferred at around
250, 500, 1500 and 1750 CE (Figure 3a), likely artefacts of higher resolution sampling in the C18 record. Beyond this, the
period between ~1300–1800 CE appears to have experienced higher I:E compared to the preceding millennium, with a possible
shift towards drier conditions after ~1800 CE (Figure 3a). The timing of this wetter phase corresponds with the Little Ice Age,
a period of significant global climate change including cooling over the Northern Hemisphere continents (Gebbie and Huybers,
2019). Several sites in south-eastern Australia have also recorded climates with high effective moisture during this period.
Lakes in western Victoria showed reduced variability with high inferred precipitation:evaporation prior to 1840 CE (Jones et
al., 2001; Tibby et al., 2018; Tyler et al., 2015). At Lake Surprise, the period is marked by a prolonged phase of low diatom-
inferred conductivity (Figure 4; Barr et al., 2014) and higher lake levels are inferred for Lake Keilambete after 1500 CE
(Wilkins et al., 2013). Furthermore, oxygen isotope ratios in the planktonic foraminifer species *Globigerina bulloides* from
sediments in the Murray Canyons offshore South Australia indicate that the Little Ice Age was possibly a prominent cold phase
(Figure 4; Moros et al., 2009).

A recent study examining coherent variability in proxy records of south-eastern Australia hydroclimate during the last 1200
years has identified two vectors interpreted to reflect hydroclimate changes (Dixon et al., 2019). The statistical analysis
included eight proxy records which met stringent criteria for inclusion in the PAGES Aus2k database (Dixon et al. 2017). The
analysis suggests an increase in effective moisture between 900 and 1750 CE and generally wetter conditions between 1400
and 1750 CE (Dixon et al., 2019). The agreement of the results of this regional synthesis with the interpretation of The Coorong
oxygen isotope record suggests that hydroclimate variability in the South Lagoon may have been consistent with broader,
regional scale variability and climate forcing. This outcome is not necessarily surprising, given the size and geographic range
of the Murray-Darling Basin (Gingele et al., 2007).



## 4.2 Drivers of regional hydroclimate in The Coorong and south-eastern Australia

Rainfall in south-eastern Australia is driven by multiple large-scale ocean-atmosphere interactions including ENSO, IOD and SAM which can affect autumn, summer and spring rainfall on different timescales (Gallant et al., 2012; Risbey et al., 2009). As such, the moisture balance in The Coorong is controlled by multiple and co-varying drivers and so is difficult to attribute to a single climate process. Of those drivers, the effect of ENSO on hydroclimate in south-eastern Australia during the last two millennia presents a conundrum, since the transition from the Medieval Climate Anomaly to Little Ice Age in the Northern Hemisphere (e.g. Wanamaker et al., 2012) is often inferred to have coincided with a shift from a more La Niña-like climate towards El Niño dominance (Mann et al., 2009; Perner et al., 2018). In south-eastern Australia, however, this shift is one towards wetter climate during the Little Ice Age in agreement with records from the western pacific (Emile-Geay et al., 2013), thus contrasting with the expected relationship between El Niño-like climates and drought (Wang and Hendon, 2007).

In a recent review of palaeoclimate informed IOD trends, variability and interactions, Abram et al. (2020b) propose that changes in mean hydroclimate from prolonged periods of drier or wetter than average conditions could be explained by changes in the strength of interannual IOD and ENSO variability rather than changes in the mean state of either forcing. Furthermore, studies of recent droughts in Australia have concluded that the lack of La Niña and negative IOD events, hence reduced interannual variability, is responsible for prolonged drought conditions (King et al., 2020; Ummenhofer et al., 2009). The shift from dry to wet conditions observed in the South Lagoon of The Coorong and other south-eastern Australian records around 1300 CE broadly coincides with mid-millennium increase in the variability of IOD and ENSO in the eastern equatorial Indian Ocean (Abram et al., 2020a), and amplified ENSO variability in the tropical Pacific Ocean (Rustic et al., 2015). Furthermore, low ENSO variability from 0–1000 CE (Tardif et al., 2019) corresponds with mostly dry conditions in the South Lagoon of The Coorong and in south-eastern Australia.

The southern westerly winds (SWW), which are largely described by the SAM index, also impact hydroclimate in south-eastern Australia (Marshall, 2003; Risbey et al., 2009), particularly the generation of frontal rainfall during the winter, albeit over shorter timescales than ENSO and the IOD (King et al., 2020). Positive phases of SAM, in association with a southerly shift of the SWW, have been related to reduced rainfall in south-eastern Australia in several studies (Cai and Cowan, 2006; Hendon et al., 2007; Risbey et al., 2009). Proxy data based reconstructions of SAM show a negative trend from ~1300 CE (Abram et al., 2014; Dätwyler et al., 2018) coinciding with the shift from dry to wet hydroclimate in the South Lagoon (Figure 4). This observation contrasts with inferred period of dry conditions recorded by pollen and charcoal data from lake sediments in Tasmania however, which are interpreted to reflect a positive SAM anomaly between 950–1450 CE (Fletcher et al., 2018). In contrast, this period is inferred to have been a period of both wet (~1000–1200 CE) and dry (1200–1300) conditions in the South Lagoon (Figure 4). It is possible that the timing of positive SAM events could cause this variation. A high index polarity of SAM during winter has been found to result in decreased daily rainfall over southeast and southwest Australia, whereas during summer it is associated with decreased rainfall in western Tasmania but increased rainfall in the south-eastern region of mainland Australia (Hendon et al., 2007). As a consequence, differences in the seasonal effects of SAM on the climate of



mainland Australia and Tasmania may explain the apparent differences in timing of wet and dry periods between the two regions. Reconstructions of SAM during the last millennium suggest increased Austral summer anomalies at ~1150 CE, which

were not evident in annual SAM anomalies which instead peak at ~1300 CE (Figure 4; Dätwyler et al., 2018). The temporal similarities in summer and annual positive SAM events and the wet and dry phases in the C18 record suggest that the season of SAM anomalies may therefore be an influential driver of multi-decadal to centennial scale hydroclimate variability in the South Lagoon, and possibly over regional south-eastern Australia.

**4.4 The pre-impacted hydrology of the South Lagoon: implications for restoration targets**

The ecological demise of the South Lagoon has led to calls for drastic action to reduce salinity and nutrient loading in the lagoon (Department for Environment and Water, 2019). The solutions considered include improvements to the connectivity between the North and South Lagoons, and maintaining the Murray Mouth to provide greater sea-water exchange with The

Coorong (Department for Environment and Water, 2019). Such proposed actions highlight an urgent need to understand the pre-impacted hydrological state of The Coorong, which in turn led to the development of this unique and iconic ecosystem. The ubiquitous presence of *A. helmsi* in core C18, and in general in South Lagoon sediments, suggests that the lagoon provided favourable conditions for this species prior to the 1950s (salinity <55 Wells and Threlfall, 1982). This finding is in agreement with the evidence that the aquatic plant *Ruppia megacarpa,* which has a salinity tolerance of 2 to 50, was the dominant species

of *Ruppia* in the South Lagoon for the 3000 years prior to European impact (Dick et al., 2011). In addition, analyses of foraminifera (Dick et al., 2011) and diatom assemblages (McKirdy et al., 2010) suggest that water salinity in the South Lagoon was relatively stable for at least 3000 years prior to the 1950s. Estimates of the pre-European salinity of the South Lagoon from microfossil and geochemical evidence range from estuarine (~5-30) (Dick et al., 2011) to slightly above that of seawater (>35) (Reeves et al., 2015) based on microfauna and microflora fossil assemblages. Previous work has highlighted the effect

of $20^{th}$ century hydrological alteration, land clearance and water abstraction in the MDB upon salinity and ecological conditions in The Coorong (Fluin et al., 2007; Krull et al., 2009). It is during this period that *A. helmsi* disappear from core C18 (~1965) and a black, clay-rich monosulfidic ooze appears - the result of anoxia and highly reducing conditions (Figure 2; Krull et al., 2009). The combination of high salinity and anoxia in the South Lagoon made conditions unsuitable for *A. helmsi* (Dittmann et al., 2019) and this, along with other environmental indicators, suggests a departure from the conditions that prevailed prior

to this time (Kingsford et al., 2011). Despite the loss of *A. helmsi* shells from the sediment core, it appears that the water balance in The Coorong has remained relatively stable for the last ~1750 years. The oxygen isotope values measured from sedimentary shells do not significantly differ from values estimated for modern populations calculated from measured water temperature and oxygen isotope composition in the South Lagoon (Figure 5a). While the range of the predicted modern values is broader for both Parnka Point

and Jack Point than those observed in core C18, the overlap in values suggests that the South Lagoon had been an isolated, highly evaporative system for the last ~1750 years. In addition to falling within the range of predicted South Lagoon



populations, the $\delta^{18}O_s$ values from core C18 are statistically different from the $\delta^{18}O_s$ values measured from modern populations in the North Lagoon (Figure 5b), which is subject to a greater influence of seawater and river water in the past and in the modern system. The waters in which the shells from the North Lagoon were collected range in salinity from 0 to 46 and in

$\delta^{18}O$ from -2.81 ‰ to 3.87 ‰ (Chamberlayne et al., 2021). The higher $\delta^{18}O_s$ values from core C18 compared to those from the modern North Lagoon suggest the South Lagoon has for the past 1750 years been a more isolated and evaporated system compared to the contemporary North Lagoon. Evidence of the North and South Lagoons being separate geochemical systems through time has also been inferred through the analysis of carbon and nitrogen isotope ratios from sediment cores from the two lagoons (McKirdy et al., 2010). Furthermore, microfauna assemblages point to the South Lagoon being more saline than

the North Lagoon for at least the last millennium (Reeves et al., 2015).

While it is evident that the South Lagoon has undergone an increase in salinity over the last century, a consequence of both climate change and direct human impact, it appears that the hydrological balance of the lagoon has been relatively constant, and that additional stressors – namely eutrophication and anoxia – may have played a major role in the recent loss of biodiversity in the system. A relatively constant hydrological balance in concert with an increase in salinity also implies that

the salinity of inflowing water to the South Lagoon must have changed. Such changes may have arisen due to an increase in the flux of saline groundwater, either directly into the South Lagoon along its margins (Haese et al., 2009) or into the Salt Creek tributary, which flows into the South Lagoon. While the flow of River Murray water into Lake Alexandrina and The Coorong may have been relatively unchanged prior to human impacts, the salinity of inflowing waters has increased since that time (Hart et al., 2020), which when subjected to intense evaporation may have enhanced the hypersalinity of the modern

South Lagoon. This study did not analyse $\delta^{18}O$ in *A. helmsi* shells in sediments of the North Lagoon, though this would allow a comparison of the hydrology of each lagoon through time and therefore be an interesting avenue for future research.

## 4 Conclusions

This study examined hydrological change over the last 1750 years in The Coorong South Lagoon, South Australia, through the analysis of oxygen isotopes of bivalve shells preserved in lagoonal sediments. Variability in $\delta^{18}O_s$ was interpreted as an

inverse function of the balance of inflow over evaporation in the South Lagoon catchment. Two extended periods of low and high inflow:evaporation were inferred from ~500 to 1050 and ~1300 to 1800 CE respectively, consistent with other hydroclimate records in the region, in turn suggesting a common driver for multi-decadal to centennial scale hydroclimate variability in south-eastern Australia. Comparison with reconstructed indices for major ocean-atmosphere interactions suggest the importance of the timing of SAM anomalies on hydroclimate in The Coorong, while changes in the variability of IOD and

ENSO may have contributed to changes in mean hydrological state. The overall range of $\delta^{18}O_s$ values measured from shells of *A. helmsi* suggest that the modern-day hydrological balance of the South Lagoon is not markedly different to that inferred for the last 1750 years, indicating the southern basin was always a highly evaporated water body. Furthermore, the sedimentary $\delta^{18}O_s$ values differ significantly from those in the modern-day North Lagoon, reinforcing previous suggestions that the two

lagoons have evolved as separate systems and that the modern North Lagoon is not an analogue for the pre-impacted South

Lagoon. Consequently, this study contributes both to our understanding of the mechanisms of natural hydrological variability in south-eastern Australia, and to our knowledge of the pre-impacted state of the South Lagoon in particular, and the ecologically significant Coorong Lagoon system generally. Such data can assist in understanding the resilience of hydrological systems to long term climate change trajectories.

**Data availability**

The original data used in this research study are archived in the Mendeley Data Repository (Chamberlayne, (2022), "1750 years of hydrological change in southern Australia: a bivalve oxygen isotope record from the Coorong Lagoon", Mendeley Data, V1, doi: 10.17632/4vktdpckhv.1)

**Author Contribution**

BKC, JJT and BMG conceptualized this study; BKC collected, curated and analysed isotope data; DH and JT collected the sediment core; JT acquired financial support for coring and radiocarbon analyses; YS and BKC acquired funding for further radiocarbon analyses; BKC visualised the data and wrote the original draft; all co-authors reviewed and edited the manuscript.

**Competing Interests**

The authors declare that they have no conflict of interest.

**Acknowledgements**

We acknowledge The Coorong, Lower Lakes and Murray Mouth region and surrounding areas as Ngarrindjeri Country and
thank the Ngarrindjeri Regional Authority for permission to undertake the coring. We would like to thank Mark Rollog for his assistance with stable isotope analyses. This project was undertaken while BKC and YS were recipients of an Australian Postgraduate Award and Australian Graduate Research Scholarship, respectively. Coring and radiocarbon dating at Waikato University obtained by DH was funded by Australian Research Council Project LP0667819. Radiocarbon dating at the Australian Nuclear Science and Technology Organisation (ANSTO) was funded by an Australian Institute of Nuclear Science
and Engineering (AINSE) Honours Scholarship to BC and an Australian National Collaborative Research Infrastructure Strategy (NCRIS) grant to YS (ANSTO portal No. 12390).

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





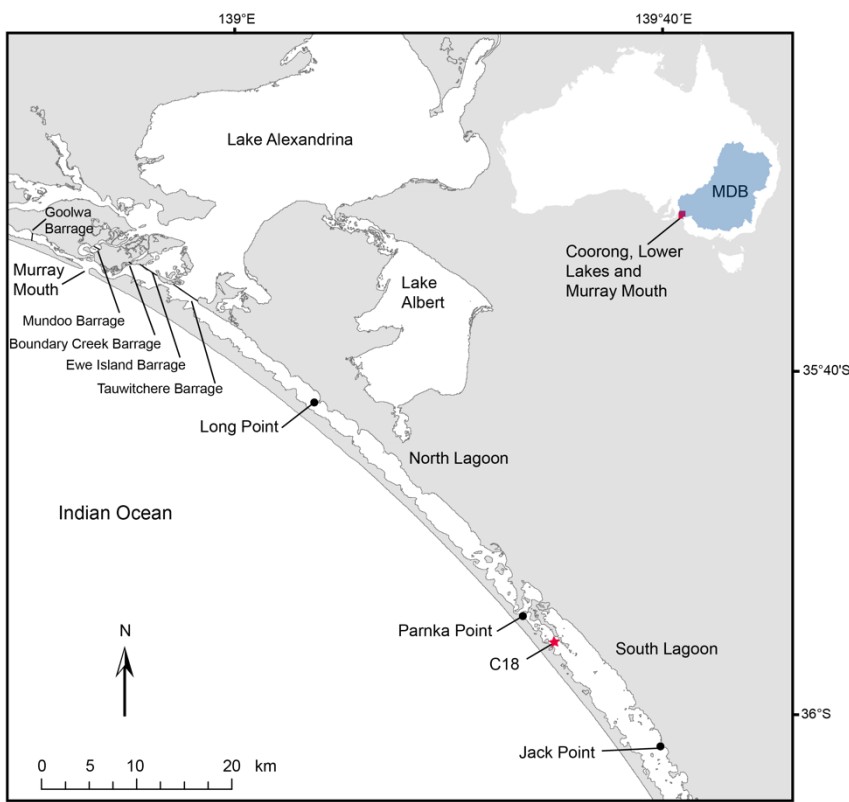

**Figure 1: Map of The Coorong, Lower Lakes and Murray Mouth – the wetland at the terminus of the Murray-Darling River system in the Murray-Darling Basin (MDB) highlighted in blue. The location for core C18 is highlighted with a red star.**



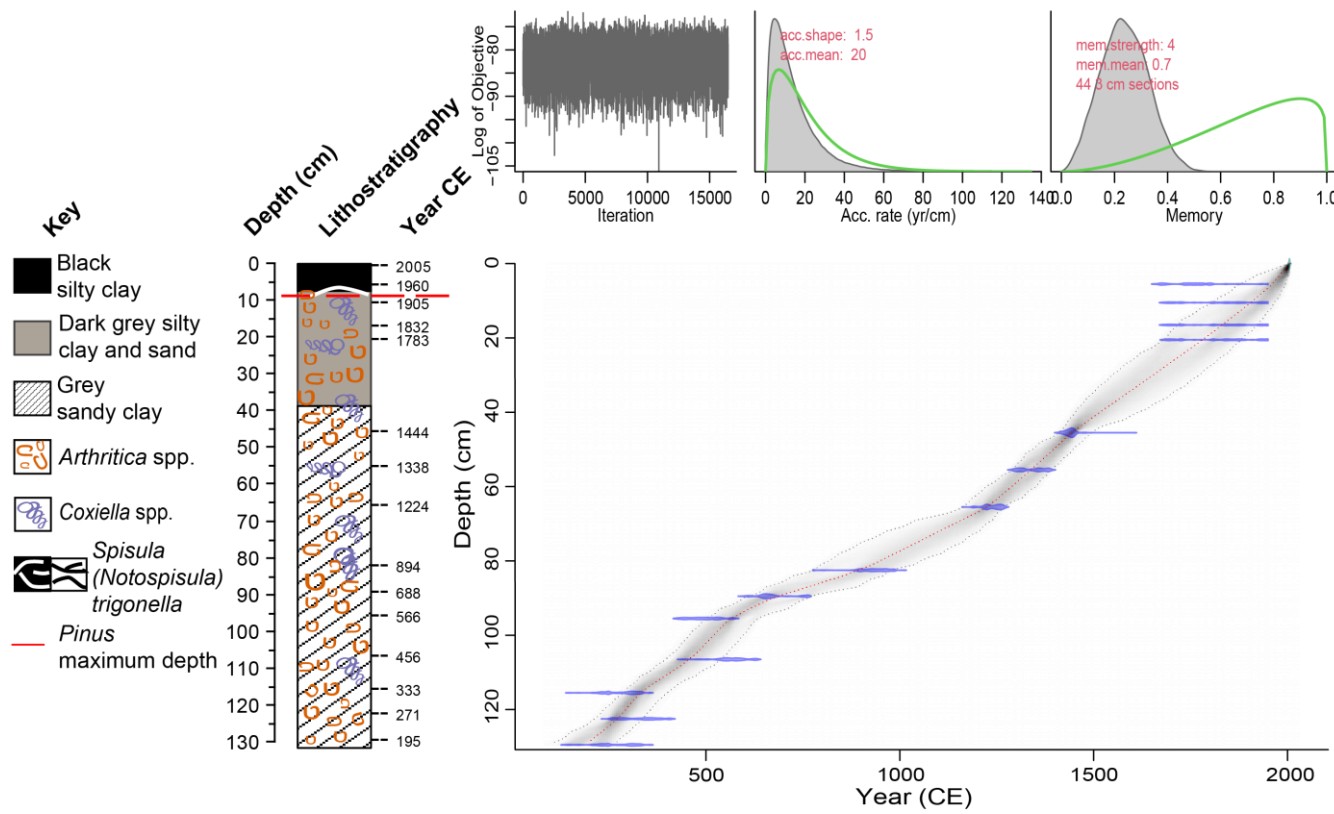

**Figure 2: Age-depth model and lithostratigraphy for core C18. Blue markers show the probability distributions for each dated sample. Grey dotted lines indicate modelled 95% confidence intervals, where darker shading indicate more likely ages. The modelled weighted mean age is shown by the red dotted line. Upper panels indicate the iterations performed, accumulation rate and memory used to construct the model.**




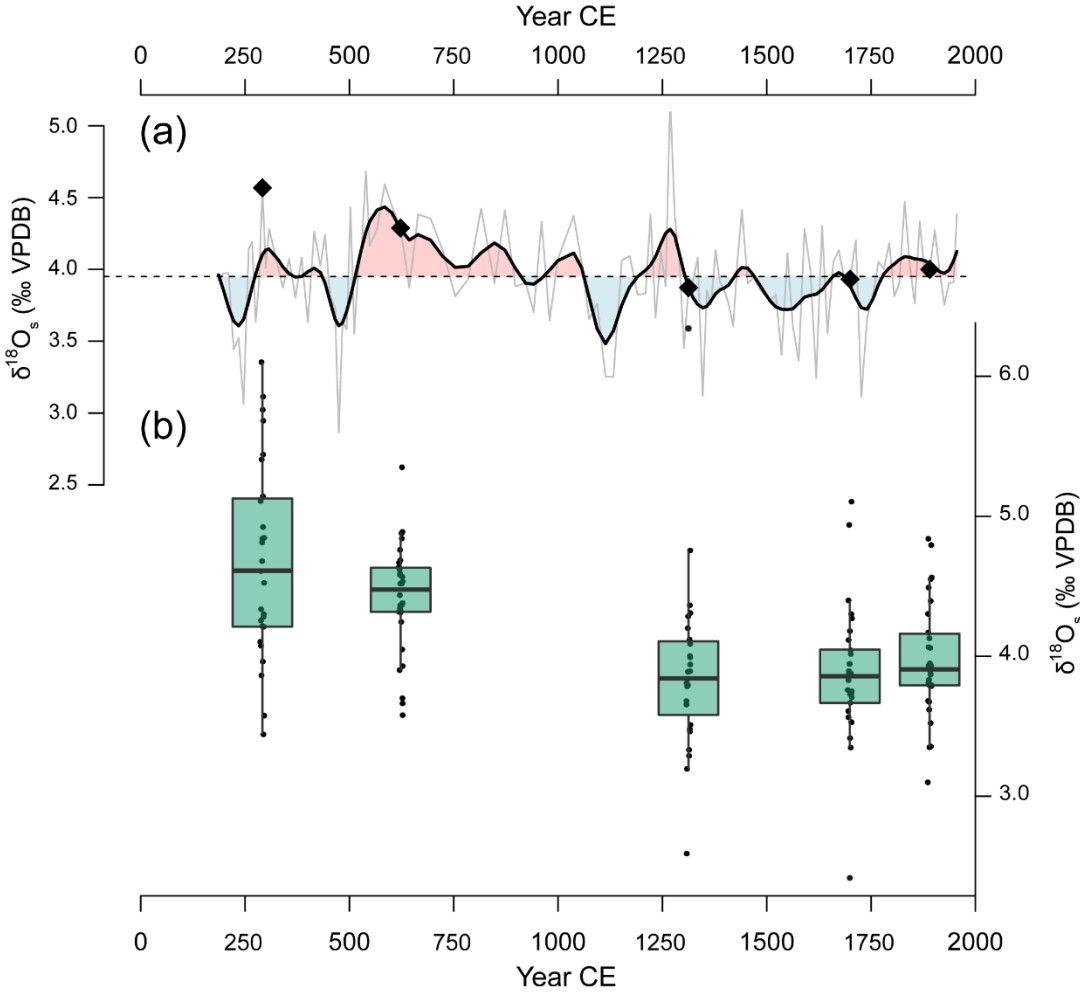

**Figure 3:** *Arthritica helmsi* **oxygen isotope ratios from core C18 plotted against time. In (a) the grey line represents a bulk measurement (n = 5) from each centimetre of C18. The dashed line indicates the mean of all values and the bold line is a smooth spline (spar = 0.4). Blue shading indicates periods of below mean $\delta^{18}O_s$, while the red shading indicates periods of above mean $\delta^{18}O_s$. Black diamonds indicate where data from multiple individuals (n ≥ 27) are incorporated into the bulk curve. (b) Boxplots of oxygen isotope ratios of *A. helmsi* shells. Data are obtained from analyses of multiple individual shells from depths corresponding to age 300 (n = 30), 631 (n = 29), 1321 (n = 27), 1708 (n = 30) and 1899 years CE (n = 27).**

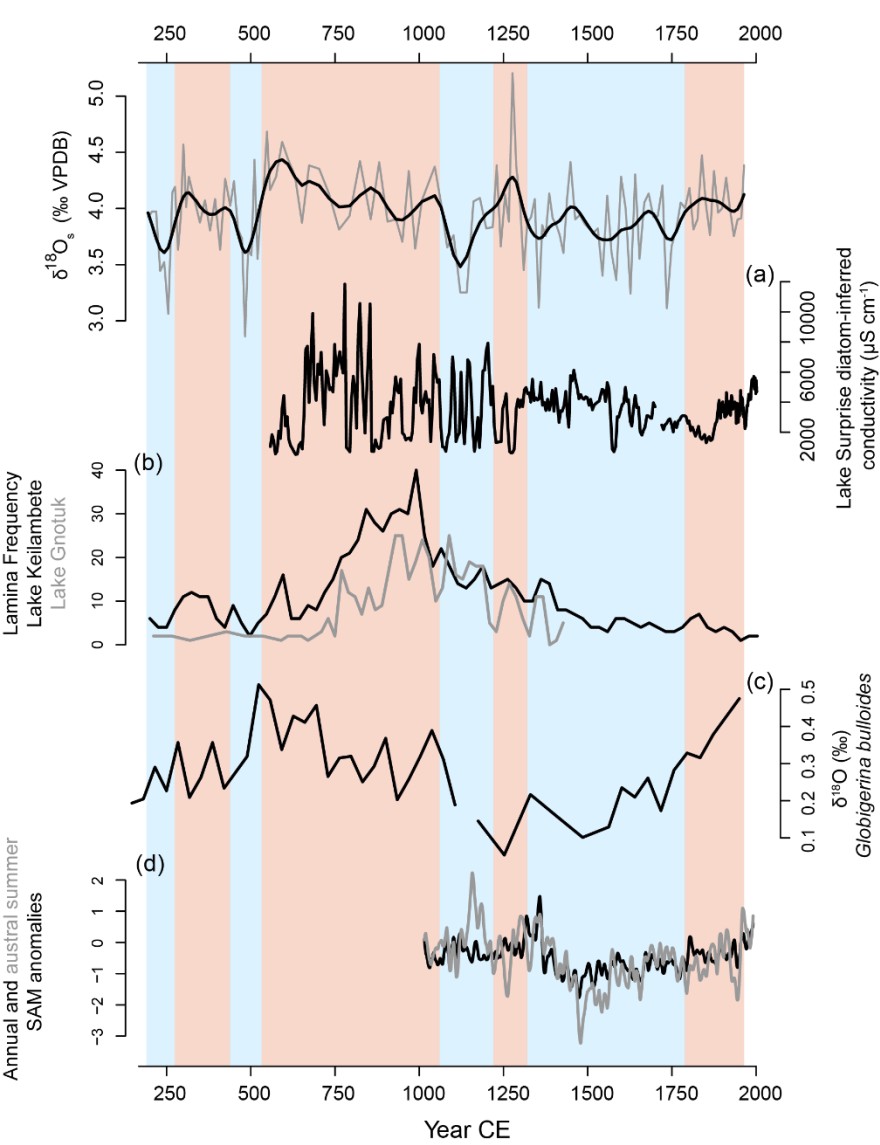

**Figure 4: δ¹⁸Oₛ from core C18 *A. helmsi* shells compared against: (a) Diatom-inferred lake conductivity from Lake Surprise, SEA (Barr et al., 2014); (b) lamina frequency in Lakes Keilambete and Gnotuk, SEA (Wilkins et al., 2013); (c) Murray Canyon core MD03-2611 δ¹⁸O *Globigerina bulloides* (Moros et al., 2009); (d) Austral summer and annual mean SAM index reconstructions (Dätwyler et al., 2018). Red and blue shading represents periods of higher and lower than average *A. helmsi* δ¹⁸Oₛ respectively.**





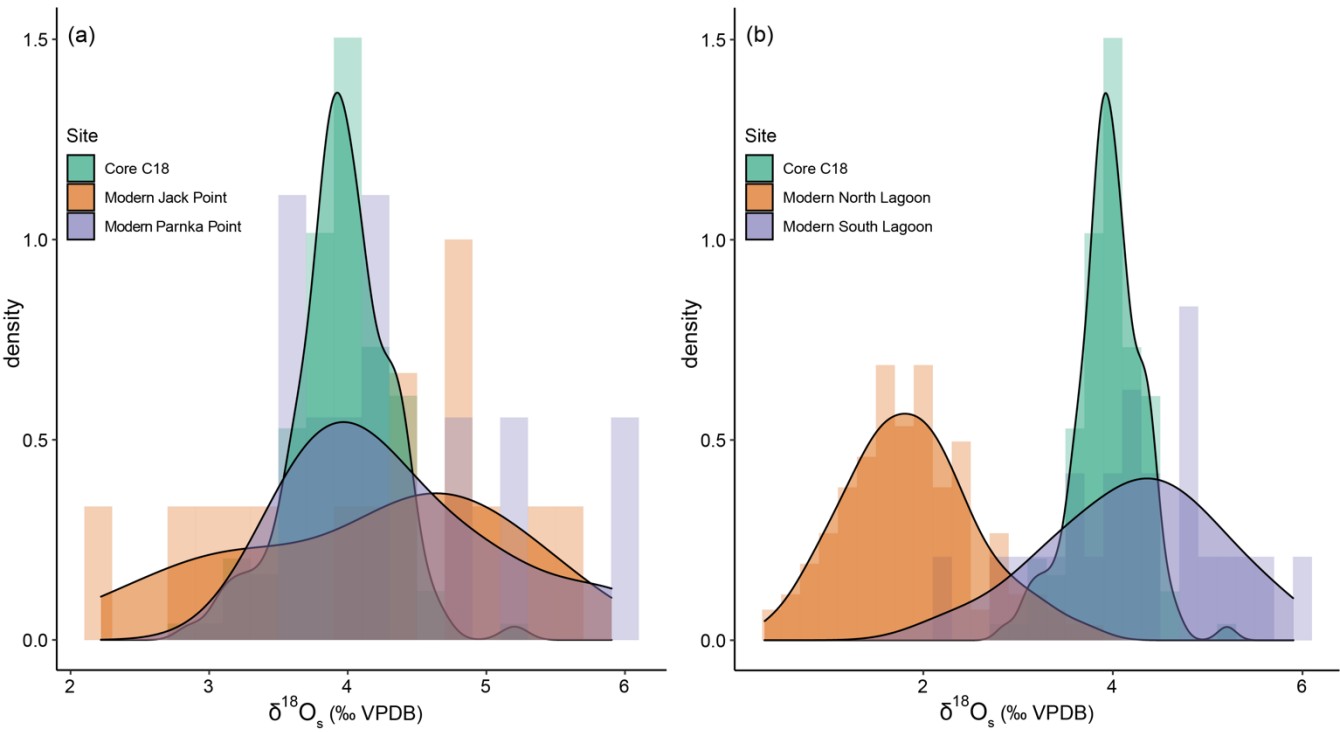

**Figure 5: Density plots with histograms of *A. helmsi* $\delta^{18}O_s$ comparing values from core C18 with (a) modern locations in the South Lagoon and (b) the North and South Lagoon. Modern $\delta^{18}O_s$ values for the North Lagoon are sourced from Chamberlayne et al. (2021). $\delta^{18}O_s$ values for the modern South Lagoon sites Jack Point and Parnka Point are calculated from temperature and $\delta^{18}O_w$ values from Chamberlayne et al. (2021) using the oxygen isotope fractionation equation developed in that study.**