# Peer review of "1750 years of hydrological change in southern Australia: a bivalve oxygen isotope record from the Coorong Lagoon"

_Climate of the Past, 2022_

## Referee Comment (RC1)

**cp-2022-39**

**1750 years of hydrological change in southern Australia: a bivalve oxygen isotope record from the Coorong Lagoon**

Briony Kate Chamberlayne, Jonathan James Tyler, Deborah Haynes, Yuexiao Shao, John Tibby, and Bronwyn May Gillanders

Review

General Comments

This paper is one in a series by the authors which look at the use of the bivalve *Arthritica helmsi* as an indicator of past hydrologic and climate change. It builds upon previous papers which look at modern and museum specimens of this species and its suitability for isotopic studies. This study is located in an area of significant interest in Australia, with regards water management, cultural connections and understanding past hydrologic conditions. The study is well-constructed, and the findings clearly presented. The establishment of the local reservoir effect for radiocarbon dating is particularly to be commended. The results are of significance not only in addressing a long-running debate in the palaeohydrology of the Coorong Lagoon, but also in adding another high-resolution climate record for southeast Australia over the past ~2000 years, allowing more robust comparisons regionally and globally.

Specific comments

Line 12: advise giving some context to the conservation and restoration efforts and how palaeoclimatic records can be useful in addressing these.

Line 30: the reference to arid and semi-arid regions here seems a little out of place. Perhaps set the geographic context of the study first.

Line 39-40: where are these located in relation to the study site – geographically and climatically? Would you expect these to be congruent?

Line 130-1 and Figure 3: You are to be commended for do analysis on individual valves as well as the bulk samples. However, the individual valves show a very broad distribution in oxygen isotope values, at any given depth. What is the justification for 5 valves per sample being representative?

Line 210: relatively dry in the context of the record or in comparison with other areas?

Line 239: Perhaps show on a map where the records in Dixon et al, 2019 are in relation to the current record. Climatically, would you expect the same response or not?

Line 335-337: I would recommend expanding on this correlation a little. How are you defining the region here and how do each of the drivers you mention here relate to wetter or drier conditions? Can you unpick the influences of each of these with respect to the variability in your record – and the regional context? How does it help build the story?

Conclusions: Suggest splitting into two sections – Firstly the palaeoclimate, and then the Coorong and management implications as separate paragraphs.

Figure 1: I would suggest including a map showing major climatic zones or influences of major climatic drivers. May also be worth including a map showing the locality of this site in relation to others mentioned in the text – both in SE Australia and globally.

Figure 4: Consider annotating which of these records were utilised in the Dixon et al., 2019 compilation.

Editorial Questions

1. Does the paper address relevant scientific questions within the scope of CP?

   Yes – a high resolution palaoeclimate/hydrology record for a region of significance

2. Does the paper present novel concepts, ideas, tools, or data?

   Yes – new taxa for isotopic records, new high resolution site to contribute to the palaeoclimate story of SE Australia and calculated radiocarbon reservoir effect for the Coorong

3. Are substantial conclusions reached?

   Yes – substantiated correlations with other regional records, attributed to climatic drivers, however with local effects recognised

4. Are the scientific methods and assumptions valid and clearly outlined?

   Yes, although I would prefer the justification for number of individuals in the bulk isotopic samples to be outlined a little more explicitly

5. Are the results sufficient to support the interpretations and conclusions?

   Yes, the interpretations and conclusions are sound – and could perhaps be a little bolder

6. Is the description of experiments and calculations sufficiently complete and precise to allow their reproduction by fellow scientists (traceability of results)?

   Yes

7. Do the authors give proper credit to related work and clearly indicate their own new/original contribution?

   Yes

8. Does the title clearly reflect the contents of the paper?

   Yes

9. Does the abstract provide a concise and complete summary?

   Yes – although there are some general statements at the start that could be contextualised more effectively

10. Is the overall presentation well structured and clear?

    Yes

11. Is the language fluent and precise?

    Yes

12. Are mathematical formulae, symbols, abbreviations, and units correctly defined and used?

   Yes

13. Should any parts of the paper (text, formulae, figures, tables) be clarified, reduced, combined, or eliminated?

   As above

14. Are the number and quality of references appropriate?

   Yes

15. Is the amount and quality of supplementary material appropriate?

   Yes

---

## Referee Comment (RC2)

[referee-annotated manuscript omitted]

---

## Author Response (AR1)

**Reviewer 1**

This paper is one in a series by the authors which look at the use of the bivalve Arthritica helmsi as an indicator of past hydrologic and climate change. It builds upon previous papers which look at modern and museum specimens of this species and its suitability for isotopic studies. This study is located in an area of significant interest in Australia, with regards water management, cultural connections and understanding past hydrologic conditions. The study is well-constructed, and the findings clearly presented. The establishment of the local reservoir effect for radiocarbon dating is particularly to be commended. The results are of significance not only in addressing a long-running debate in the palaeohydrology of the Coorong Lagoon, but also in adding another high-resolution climate record for southeast Australia over the past ~2000 years, allowing more robust comparisons regionally and globally.

Specific comments
Line 12: advise giving some context to the conservation and restoration efforts and how palaeoclimatic records can be useful in addressing these.

R: Added "In addition, these data can provide a temporal perspective to setting realistic benchmarks for ecological management not provided by instrumental data (Saunders and Taffs, 2009)." To the Introduction.

Line 30: the reference to arid and semi-arid regions here seems a little out of place. Perhaps set the geographic context of the study first.

R: We have deleted the reference to separate regions to simplify the text.

Line 39-40: where are these located in relation to the study site – geographically and climatically? Would you expect these to be congruent?

R: We have added to Figure 1 a map showing the locations of the study sites from these studies in relation to the site in the current study. The sites are from southeast Australia so

the authors would expect that the locations are influenced by the similar climate drivers as the site in the present study. A map illustrating the proximity of records is a good addition to the manuscript – thank you for the suggestion.

Line 130-1 and Figure 3: You are to be commended for do analysis on individual valves as well as the bulk samples. However, the individual valves show a very broad distribution in oxygen isotope values, at any given depth. What is the justification for 5 valves per sample being representative?

R: Thank you for this comment, we agree this information is important and should be added to the text. Five valves was selected as it meant there would be consistency across the study as the number of valves in each sediment layer was variable. It was also found in the lead authors Honours Thesis (Chamberlayne, 2015) that five valves was representative of the trace element value in a larger sample size of 16 valves. We added "Five valves was found to be representative of trace element variability in a larger sample (Chamberlayne, 2015) and allowed the sample size to remain consistent throughout the core." To section 2.4.

Line 210: relatively dry in the context of the record or in comparison with other areas?

R: The authors are stating that it is relatively dry in the context of the record and added this to the text in section 4.2.

Line 239: Perhaps show on a map where the records in Dixon et al, 2019 are in relation to the current record. Climatically, would you expect the same response or not?

R: We have added to figure 1 a map showing the locations of the study sites from these studies in relation to the site in the current study. The sites are from southeast Australia so the authors would expect that the locations are influenced by the similar climate drivers as the site in the present study.

Line 335-337: I would recommend expanding on this correlation a little. How are you defining the region here and how do each of the drivers you mention here relate to wetter

or drier conditions? Can you unpick the influences of each of these with respect to the variability in your record – and the regional context? How does it help build the story?

R: Clarified the response to SAM in the Coroong.

Conclusions: Suggest splitting into two sections – Firstly the palaeoclimate, and then the Coorong and management implications as separate paragraphs.

R: The authors have split the conclusion into two sections for clarity of ideas.

Figure 1: I would suggest including a map showing major climatic zones or influences of major climatic drivers. May also be worth including a map showing the locality of this site in relation to others mentioned in the text – both in SE Australia and globally.

R: The authors edited Figure 1 to include the location for other proxy records mentioned in the text. We decided against adding climate drivers to this map to keep it streamlined.

Figure 4: Consider annotating which of these records were utilised in the Dixon et al., 2019 compilation

R: The authors have added this information to the figure caption.

**References:**

Chamberlayne, B., 2015. Late Holocene seasonal and multicentennial hydroclimate variability in the Coorong Lagoon, South Australia: evidence from stable isotopes and trace element profiles of bivalve molluscs, honours thesis, University of Adelaide, Adelaide.

**Reviewer 2**

I enjoyed reading the paper which aims at addressing 2 important scientific questions.

- The last 2 millennia of climate change in SE Australia

- Address issues that have so far been controversial concerning the 'health' of The Coorong which has recently been modified by human activities and perhaps also as a result of 'climate change'., This is a Ramsar site of great importance and therefore deserves better understanding with respect to its past, present and future.

I have placed a number of comments directly on the manuscript.

Many are trivial such as the need to hyphenate and place comas, but towards the end of the manuscript, especially for some figures changes are necessary. The correlation with the crater maar lake records of western Victoria need changing and more importantly the comparison with the marine cores [2611 and MUC3] need changing. The record by Perner [as referred to in the manuscript] does not cover the period mentioned in the manuscript and needs to be replaced. The data by De Deckker et al. (2020) are available at the Pangaea.de web site. I will send the relevant data to the corresponding author as it was not possible to attach more than one file.

I also disagree that the Little Ice Age was not discussed by those authors who dealt with core records in Australia. See my comments using stickies.

Overall, this is an important study that needs to be published but only after amendments. I ticked the box recommending major revision. I would have preferred to tick a box saying 'moderate revision' as the suggested changes concern figure 3 especially and comments on some of the features in it that figure are not discussed in the paper and, as yet, they are very important.

The authors have commented directly on the PDF supplied by the reviewer.

The record in Figure 4 that the reviewer is referring to is from Moros (2009) not Perner as suggested in the above comment. We have though separated the record into the two cores which cover separate time periods.

**Reviewer 3**

The two primary concerns I have are 1) if this is a temperature or hydroclimate record and 2) in relation to the age model.

Firstly – the authors provide fairly weak evidence that despite establishing a d18O and temperature relationship for this site, they believe their d180 record better represents rainfall/flow. The authors need to provide further evidence/discussion of this through analysis between contemporary flow/rainfall and d180 in the top layers. This is crucial as an

alternative interpretation of the results (based on the stated temperature relationships) is that that 500-1100 was cool rather than wet and 1100-1750 was warm rather than dry.

R: The authors thank the reviewer for this valid comment, but we do not believe it would be possible to add to this study due to the lack of samples in the top centimetres of the core from the South Lagoon. Furthermore, while temperature may contribute to changes in d18O, the change required to result in the range of d18O values measured in A. helmsi valves is unrealistic for the region (~10 °C). This is discussed in section 4.1 of the manuscript.

Secondly – The top 500 years of the age model doesn't appear to be well constrained and the authors exclude some dates based on being outliers. There is a sedimentary horizon at 40cm with a date below of 1444 and a date above (20cm) of 1783. This is a big gap. Could there be a hiatus in between these dates or a change in deposition rate? Another question is if the shells were in dead or live position as I am aware they can burrow into sediments and therefore may 'move into older layers'. Also the marine reservoir effect may vary over time. For example, a bushfire could result in a high influx of young carbon, so subtracting 800 years in this instance would be erroneous. If the lake has a lot of vegetation and organic matter surrounding it, this may act to offset the carbon - groundwater age. While this may not be able to be completely addressed, the authors should discuss this in their discussion and conclusions as potential sources of uncertainty.

R: The points made by the reviewer in this comment are valid and we agree that a more detailed discussion of these potential uncertainties should in included in the text. Resolving these uncertainties is outside the scope of this study, but the authors will highlight the potential for further research to better constrain the timing of changes in the d18O record. We added a paragraph to section 4.2 addressing these uncertainties and also added the suggestion to replicate this study on other sediment cores from the Coorong Lagoons to the Conclusions.

Other comments:

In the first line of the abstract, the authors mention the resilience of aquatic ecosystems, however, palaeoclimate data is relevant to all ecosystems including terrestrial.

R: We have removed the word aquatic

Line 35 – highlights the lack of decadal scale records. However this statement is true for both high and low frequency reconstructions Reference to Dixon et al 2007 as 'recent' is probably not quite right. There have been advancements in the last 5 years. Particularly for Tasmania for example but also WA.

R: Dixon et al was published in 2017.

Of the 9 records in Dixon, how many are on the mainland in SE Australia – it is worth noting for context for this study.

R: We added a map to Figure 1 showing the locations of the sites for the proxy records synthesised in Dixon et al. (2019) in relation to the site in the current study.

Line 40 – I don't think it is appropriate to lump the ANZDA in with reconstructions based on a single remote proxy. The ANZDA is based 176 tree-ring chronologies and one coral series from both Australia and wider Pacific.

R: We have deleted reference to the ANZDA reconstruction.

Line 47 – A flood is not a decadal phenomenon. Floods tend to build rapidly, peak and subside within a week or so. Droughts can last seasons to years. The term pluvial or flood dominated epoch would be more appropriate.

R: Agree, we have changed this text to "drought or flood dominated epochs".

It would be good if the authors could provide a stronger case as to why low frequency reconstructions are useful. In the first line of the introduction the authors state that "Multi-decadal to centennial records of past hydroclimate variability are crucial for understanding long term climate drivers, for calibrating and validating climate models, for assessing

hydroclimate sensitivity to external drivers and for estimating the probability of multi-decadal climate extremes", however much of this requires annual or sub annual data. I am not suggesting low frequency reconstructions that cover longer periods are not useful but please spell out why they are an important piece of the puzzle in the introduction.

R: Added "These data can provide a temporal perspective to setting realistic benchmarks for ecological management not provided by instrumental data (Saunders and Taffs, 2009)." To the introduction.

Line 70 – 'Moreover' should be 'however'

R: Amended

Line 91 – Remove "formally speaking"

R: Amended

The authors mention the use of Pinus Pollen in identifying when modern section of the record. How is the pollen actually identified? Please include details

R: The authors will clarify in the text that the pollen collection and identification method is outlined in another paper (Krull et al. 2009).

Line 135 – missing "the"

R: Added

The authors mention some samples with erroneous dates, where they were identified as older than the sequence in which they were in. is this evidence of the aged carbon offset not being constant? If not, why might then be in error?

R: The erroneous dates may have been a consequence of mixing in the top sediments during storm activity or similar. Another possibility is that old shells were washed into the lagoon. Along the banks of the lagoons are past shorelines which contain abundant amounts of A. helmsi shells. It is possible that these were transported by wind, water or wildlife. We have added this information to a new paragraph at the end of section 4.2.

Figure 1 – Add a box around Australia as it is a different scale to the rest of the map

R: We separated the figure into two sections which addresses this comment.